# Associations between Degenerative Lumbar Scoliosis Structures and Pain Distribution in Adults with Chronic Low Back Pain

**DOI:** 10.3390/healthcare11162357

**Published:** 2023-08-21

**Authors:** Shoji Kojima, Tatsunori Ikemoto, Young-Chang Arai, Atsuhiko Hirasawa, Masataka Deie, Nobunori Takahashi

**Affiliations:** 1Department of Orthopaedic Surgery, Aichi Medical University, Nagakute 480-1195, Japan; koji23.go.for@gmail.com (S.K.); athkpixy10@gmail.com (A.H.); masatakadeie@yahoo.co.jp (M.D.); ntakahashi0617@aichi-med-u.ac.jp (N.T.); 2Multidisciplinary Pain Center, Aichi Medical University, Nagakute 480-1195, Japan; arainon@aichi-med-u.ac.jp; 3Department of Orthopaedic Surgery, Hiroshima Citizens Hospital, Hiroshima 730-8518, Japan

**Keywords:** low back pain, degenerative lumbar scoliosis, pain distribution

## Abstract

Background: This study aimed to investigate the location and distribution of pain in adults with chronic low back pain (LBP) with degenerative lumbar scoliosis (DLS) according to coronal deformities. Methods: We enrolled 100 adults with chronic LBP and DLS, dividing them into two groups, a right-convex DLS group (n = 50) and a left-convex DLS group (n = 50). Dominant pain location was analyzed by dividing it into three parts—left side, right side, and center—and pain areas were identified using the pain drawing method; then, a heat map was created for each group. An association between pain location and convex side was analyzed as the primary outcome. Additionally, we assessed pain characteristics and radiological parameters, such as the curve structure and degree of degeneration. We used the Mann–Whitney U test or the chi-squared test to compare the clinical characteristics of the two groups, and generalized linear models were utilized to determine which variables were associated with pain severity or pain area. Results: The results indicated that there was no significant difference between the two groups in terms of the association between the curve structure, pain severity and location. In multivariate analysis, although we did not find any variables associated with pain severity, we observed that age and a left-convex DLS were negatively correlated with pain area among all participants. The heat map demonstrated that individuals with chronic LBP frequently experienced pain in the central lumbar region, regardless of the coronal curve structure. Conclusions: Our findings suggest that degenerative coronal lumbar deformities may not have a specific pain pattern associated with a curved structure.

## 1. Introduction

Chronic low back pain (LBP) is a common health problem worldwide. Although the cause of LBP is usually unidentifiable even with imaging, degenerative lumbar scoliosis (DLS) is believed to be a cause of chronic LBP accompanied by degenerative spinal malalignment in elderly individuals [1,2,3].

The DLS is defined as a spinal deformity in a skeletally mature individual with a Cobb angle > 10° in the coronal plane and is caused by accelerated degeneration of the lumbar spine in middle-aged patients with intervertebral disc and facet joint degenerations [1,4,5]. As the pathophysiology of DLS is associated with degenerative changes, such as degenerative discs and deformities of facet joints, it was expected that there may be trends in pain location and distribution toward the curved structure if a degenerative curved structure contributes to pain location.

Prior studies have indicated correlations between the extent of coronal curvature and the severity of LBP [6,7,8], although several studies have found no significant relationship between radiographic coronal parameters and the severity of pain in individuals with DLS [9,10]. In addition, some studies suggest that pain on the convex side originates from the paraspinal muscles [11,12], while alternative research proposes that this pain can also arise from the facet joints [13]. In a finite element method (FEM) analysis of a degenerative scoliosis spine, Wang et al. observed compressive deformation of the facet joints predominantly on the concave side [14]. Pain at the concave side of the scoliotic curve is hypothesized to be due to damaged facet joints [11] and degenerative alterations in the intervertebral disc spaces [12]. These findings hint at a potential relationship between the direction of spinal deformity (left or right) and the laterality of pain. For instance, it has been postulated that individuals with right convex DLS may be more prone to experiencing low back pain on the right (or left) side. However, to our knowledge, existing studies have not investigated pain characteristics by categorizing them into left and right curvatures except one research. Berry et al. reported that out of the patients with scoliosis, four experienced only back pain symptoms: two on the concave side of the curve and two on the convex side. Furthermore, 41 patients exhibited radiating symptoms, with 20 on the concave side of the curve and 21 on the convex side [15]. This suggests that there is no evident relationship between the direction of the curve and the symptoms. Nonetheless, this study is not without its limitations, including an insufficiently detailed assessment of pain sites and a restricted sample size.

To specify pain characteristics in individuals with chronic musculoskeletal disorders, recent studies have focused on pain location and distribution among patients with hip osteoarthritis [16], knee osteoarthritis [17] or chronic LBP [18,19] using a pain drawing method [20]; however, to the best of our knowledge, no study has reported pain location and distribution in individuals with LBP associated with DLS. Therefore, this study aimed to investigate the association between location and distribution of pain and coronal lumbar deformities in adults with chronic LBP and DLS.

## 2. Materials and Methods

### 2.1. Study Design

This was a cross-sectional study, and retrospective review of medical records. The study procedures were performed in accordance with the Strengthening the Reporting of Observational Studies in Epidemiology guidelines for cross-sectional investigation [21].

Given that the protocol for this study is novel to our team, we will provide a detailed description in the subsequent paragraph.

### 2.2. Ethics and Consent Forms

All procedures performed in studies involving human participants were in accordance with the 1964 Helsinki Declaration and its later amendments or comparable ethical standards. This study was approved by the Aichi Medical University Hospital Research Ethics Board (2021-172). A comprehensive agreement for academic use of information such as type of treatments, treatment progress or any other data acquired during their treatments was obtained from the patients by the hospital at the time of their visits. The contents of this study were made public through the hospital’s website. As this study had no interventional approach, the requirement for written consent was waived for participants evaluated during the study period unless they refused to provide information in accordance with the opt-out strategy [22].

### 2.3. Eligibility

The inclusion criteria for participation in this study were as follows: (1) persistent LBP extending beyond a period of six months, (2) individuals aged 40 or above, and (3) patients diagnosed with DLS elaborated in the ensuing sections. The exclusion criteria were as follows: (1) presence or history of major neurological disorders, such as post-stroke or Parkinson’s disease, (2) ongoing malignant disease, (3) ankylosing spondylitis or pyogenic spondylitis, and (4) dementia.

### 2.4. Participants

Eligible participants in this study were recruited from the outpatient ward of the Aichi Medical University Hospital between January 2021 and February 2022. Participants were recruited according to the eligibility criteria until the target number of participants in each group was reached.

### 2.5. Assessment of Scoliosis

An anterior–posterior plane radiographic examination was performed for all participants in the standing position at the time of enrollment. The apex of the curve in the coronal plane and the apical level were determined. The apical vertebra was defined as the vertebra with the greatest distance from the midline with the most rotation. Curves with an apex between the first lumbar disc and the fourth lumbar vertebral body are considered lumbar curves [23]. The largest Cobb angle was used to determine whether the participants had DLS, and participants with a Cobb angle > 10° were classified as having scoliosis according to the Schwab criteria [10]. In addition, since degenerative scoliosis differs from adult idiopathic scoliosis in terms of the presence or absence of degenerative findings, we defined DLS as a spinal curve exceeding 10 degrees accompanied by degenerative intervertebral discs and/or vertebral osteophytes in the lumbar spine. In the DLS patients, the participants were further divided into a right-convex or a left-convex DLS group based on their curve structures.

### 2.6. Assessment of Degeneration

It is believed that asymmetric disk degeneration is the beginning of the coronal spinal changes in de novo scoliosis [4]. Additionally, a previous study reported that both disc degeneration and osteophyte formation correlate with the curve’s structure [24]. Therefore, in this study, the degree of degeneration in the lumbar spine was determined via findings of degeneration in the lumbar intervertebral disc and osteophytes just below the apical vertebra in the lumbar spine.

The schema is shown in Figure 1. The degree of degeneration of the intervertebral disc was determined by measuring the right and left sides of intervertebral heights. If the ratio of the right to left heights was 0.8 or greater, it was classified as no degeneration (level 0); if the ratio was 0.2 or greater but less than 0.8, it was classified as level 1; if the ratio was less than 0.2, it was classified as level 2. The degree of osteophytes was defined as follows: no degeneration (level 0) if the size of the osteophyte just below the apical vertebra was less than 2 mm, level 1 if it was greater than 2 mm but less than 10 mm, and level 2 if it was greater than 10 mm. Finally, a degeneration score was calculated by adding the levels of the above two degenerative findings (range: 0–4). Since 0 was determined as no degeneration, the degeneration score in the participants ranged from 1 to 4.

### 2.7. Assessment of Pain

Eligible participants were asked to depict the area of pain on a silhouette of the lumbar figure with instructions from the authors using examples (Appendix A). The spatial allocation of pain is predominantly assessed through pain drawing method, wherein the afflicted individual or participant delineates regions of their pain upon a visual representation of the human body silhouette [20]. Pain drawings are frequently utilized for the evaluation of patients with LBP and their reliability has been previously reported [18,19]. The participants were also asked to indicate which part of the lumbar region was dominantly painful in daily life: the center, right side, or left side. If the pain radiated beyond the buttocks to the lower extremities, the pain up to the buttocks was recorded as back pain.

We investigated the main pain location by dividing the area into the left side, right side, and center, and analyzed the association between pain location and convex side as the primary outcome. Specifically, we examined whether the main pain location was convex- or concave-dominant in the two DLS groups.

Second, we investigated the relationships among pain characteristics, such as pain intensity, pain area, demographics, and curve characteristics. A standardized lumbar chart printed on A4 sheets was provided, and the patients were instructed to circle every part of their body where they felt pain. The participants were asked to rate the severity of pain on a scale of 0–10, with 0 indicating no pain and 10 indicating the worst pain imaginable (NRS: numerical rating scale). Pain intensity was determined using the maximum NRS score on the lumbar chart. The size of the pain area on the lumbar chart was analyzed using Image-J^®^ (version 1.53a, National Institutes of Health, Bethesda, MD, USA). The A4-size silhouette sheet was transformed into a JPEG image and imported into Image-J^®^ software. Each individual’s pain perception area was measured using the tool provided within the software. If multiple pain areas were present, all were measured and the combined total was defined as the individual’s pain area using a standardized arbitrary unit.

Third, we created a pain heat map representing the pain location for each of the two groups. To create a pain chart, the pain area of individuals was transferred to an excel sheet consisting of 4630 pixels. Subsequently, a heatmap was created by unifying the coordinates of the pain site for each group of 50 subjects and adding the values on the coordinates on an excel sheet. (Figure 2).

### 2.8. Sample Size

Sample sizes were not estimated because we could not obtain any hypotheses from the literature. Therefore, we determined the sample size based on the following reports. A minimum of 41 participants are needed per group to detect differences in pain sensitivity [25]. Furthermore, when examining the psychological characteristics of pain, 50 participants in each group, namely intervention and non-intervention, were desirable to obtain high-quality results [26]. Therefore, the present study included 50 participants each in the right-convex DLS and left-convex DLS groups. We enrolled consecutive patients with chronic LBP until there were 50 participants in each group.

### 2.9. Statistics

Continuous variables are presented as mean and standard deviation (SD) or as medians and interquartile ranges (IQR), while categorical variables are represented as the number and percentage of patients. The Shapiro–Wilk test was used to investigate whether the data were normally distributed.

First, the association between the curve’s structure and pain site in the two DLS groups was analyzed using the chi-squared test as the primary outcome. Second, we compared the differences in clinical characteristics between the two groups using the Mann–Whitney U test or the χ^2^ test for categorical variables. Third, generalized linear models were used to determine which variables were associated with pain severity or pain area. In a generalized linear model, each outcome of the dependent variables is assumed to be generated from a particular distribution in an exponential family that includes the normal, binomial, Poisson and gamma distributions [27], and Akaike’s information criterion was used as a criterion of model selection in each model. Analyses were performed using the SPSS software (version 25, SPSS Inc., Chicago, IL, USA). All results were considered to be statistically significant at *p* < 0.05.

## 3. Results

### 3.1. Characteristics of the Participants

There were no statistical differences in the demographic background, age, or sex ratio between the two groups. In addition, pain severity and area did not show statistical differences among the two groups (Table 1).

### 3.2. Radiographic Parameter, Pain Locations and Distributions in the Participants

The majority of participants (86% in left-convex, 88% in right-convex) had mild to moderate curves with Cobb angles of less than 40 degrees. The distribution of the Cobb angle for each group is shown in Figure 3. The severity of scoliosis, degeneration score and levels of apical vertebrae were not significantly different between the right- and left-convex groups. In terms of the primary outcome, the proportion of scoliosis in which the right and left sites of pain coincided with their respective curve structure was 36% in the right-convex DLS group and 30% in the left-convex DLS group, with no significant difference in the association between the curve structure and the site of pain between the two groups (*p* = 0.67) (Table 2).

The heat maps showed that the two groups had frequent pain in the central lumbar region regardless of the coronal curve’s structure (Figure 4).

### 3.3. Predictors for Pain Intensity or Area

Finally, age, sex, curve type, and Cobb angle were used as independent variables in generalized linear model analyses for predicting pain severity or pain area. Although no variables were associated with pain severity, age and a left-convex DLS were negatively associated with pain area among participants (Table 3).

## 4. Discussion

This study investigated the location and distribution of pain in adult patients with chronic LBP and DLS. Contrary to our expectations, no significant difference was noted in the association between the curve structure and pain area in patients with DLS. Further, patients with chronic LBP had frequent pain in the central lumbar region, regardless of the coronal curve structure. To the best of our knowledge, this is the first study that examined the location and distribution of pain in adults with chronic LBP and DLS.

De novo DLS is believed to arise from the asymmetric degeneration of the disc and facet joint [4,13,15,24,28]. While idiopathic scoliosis is rarely associated with back pain in adolescents [13,29], the most frequent clinical problem in the DLS population is thought to be LBP [1,13]. Yet, the possibility of a unique pain pattern associated with the curvature in degenerative coronal imbalance remains unclear. Idiopathic scoliosis typically manifests as a spinal curvature predominantly in younger ages, while DLS emerges as a degenerative condition predominantly among elderly individuals. This latter condition is presumed to involve pathological changes in various tissues, such as intervertebral discs, facet joints, and muscles. One study indicated no correlation between unilateral symptoms and curve direction in DLS patients [15]. This prior research, however, did not delve deeply into the localization of pain sites. Addressing this void, our study employed the pain-drawing method to provide a more detailed evaluation of the patients’ pain distribution. Consistent with earlier findings, our data did not identify any clear connection between curve direction and pain location in DLS subjects, even with an expanded sample size. Moreover, our research suggested the central region of the lower back as a pain site in chronic LBP patients, regardless of deformities in the coronal plane. Our results also align with previous claims, underscoring the tenuous link between scoliosis curvature severity and pain intensity [9,10]. These results suggest that perception and site of pain in LBP patients with DLS do not estimate the pathological damages of a degenerative spine, indicating that LBP with DLS may be included as a non-specific type.

Non-specific LBP is probably the result of a combination of biological, psychological, and social factors and accounts for approximately 80–90% of all LBP cases [30,31]. If low back pain is primarily on one side, it could be due to issues in visceral organs. This type of pain is often linked to damages of organs such as the intestines, kidneys, or ureters, while lumbar spine pain is usually attributed to problems in the intervertebral discs, facet joints, paraspinal muscles or sacroiliac joints [32,33,34,35]. While the likelihood of pain originating from the intervertebral discs is considered high, pinpointing such pain presents challenges [32]. In the case of pain stemming from the facet joints, there’s a higher probability that pain would manifest on the affected side of the referred pain [34]. Noonan et al. provided an overview of the prevailing knowledge on the pathophysiological traits often observed in the paraspinal muscles in cases of chronic LBP. They highlighted that the existing literature clearly associates changes in muscle structure and function, especially fatty infiltration and fibrosis, with low back pain [35]. In this study, we focused solely on the structure of the lumbar curve and did not evaluate the nature of the muscles using MRI. Therefore, a separate study would be required for this assessment. Given our present knowledge, precisely determining the source of low back pain remains a significant challenge [36]. Some authors have reported the clinical usefulness of the pain-drawing method to evaluate patients with LBP [37,38,39]; however, most studies investigated only self-reported pain severity and not details of the pain site among that population. A recent review indicated associations between pain-drawing and psychological characteristics in patients with chronic musculoskeletal pain [40]. The authors found that depression may be associated with pain. In this study, however, psychosocial assessments were not performed; therefore, the lack of assessment of psychosocial issues is also one of the limitations of the present study.

In multivariate analyses, although pain severity did not differ significantly between the two groups, the pain area was associated with age and a curved structure. Patients with right-sided DLS had a wider area of pain than those with left-sided DLS after controlling the Cobb angle. Although a previous study reported that curves with an apex above L2 were convex to the right, whereas curves with an apex below L2 were more frequently convex to the left in a DLS population [15], our data did not show a similar trend. This discrepancy indicates that the same results in different populations are required to identify the common features in patients with DLS. Since this is the first report indicating an association between pain area and a curved structure in individuals with scoliosis, further validation studies are required to confirm this association in a different DLS population.

Interestingly, despite the small correlation coefficient, aging was significantly negatively correlated with pain area. It is well known that the nerve conduction velocity and the amplitude of the compound action potential of sensory neurons decrease with age [41]. Although numerous studies have investigated the effect of aging on pain sensitivity [42,43], studies addressing the effect of aging on the extent of pain are scarce. Pain propagation is reflected by a widespread increase in pain-related brain activities [44]; therefore, narrowing of the perceived pain area may be related to the shrinking of the brain cortex with aging [45]. This association needs to be confirmed in future studies.

Several limitations must be considered when interpreting the results of this study. Firstly, we did not consider pathologies in the sagittal plane parameter. Although we did not evaluate sagittal spinal alignment, pathologies, such as spondylolisthesis, kyphotic malalignment, or pelvic parameters may have affected the results of this study [46]. Moreover, the study did not delve into the pathology of the facet joints or paravertebral muscles, even though they could be pertinent to the manifestations of low back pain. This was primarily due to the challenges associated with discerning the extent of facet joint deformities from frontal x-ray images. We also did not evaluate the characteristics of the sacroiliac joints, notwithstanding our exclusion of spinal inflammatory conditions like ankylosing spondylitis. Secondly, the predominant portion of our participants exhibited mild to moderate curvatures, with Cobb angles below 40 degrees. The trends might differ if the sample was more confined to severe curve deformations, such as those with a significant coronal imbalance [6]. Thirdly, the sample size in our study was relatively small. A study with a larger sample might have yielded different results. Nonetheless, it is essential to note that no matter by how much the sample size is increased, it is impossible to conclusively prove that there is absolutely no difference. Assuming a minute difference exists and given an effect size d of 0.1, an α error of 0.05, and a power of 1 − β = 0.8, the required sample size would be 39,241 cases for each group [47]. Consequently, a global-scale study might be warranted. Fourthly, as discussed above, psychological issues may be associated with the results from pain drawings. Fifthly, although past research showed that the pain-drawing method in individuals with chronic LBP is reliable, this study did not investigate the reliability of pain drawings among this study population. If the location and area of pain in patients with DLS vary over time, more complex components need to be understood to determine the causes of LBP in patients with DLS.

## 5. Conclusions

This study investigated the location and distribution of pain in adult patients with chronic LBP and DLS. Contrary to our expectations, no significant difference was noted in the association between the curve structure and pain location in patients with DLS. Further, patients with chronic LBP had frequent pain in the central lumbar region, regardless of the coronal curve structure. This study highlights the complexity of diagnosing the causes of LBP in patients with DLS and suggests that further research is needed to confirm these findings in different populations.

## Figures and Tables

**Figure 1 healthcare-11-02357-f001:**
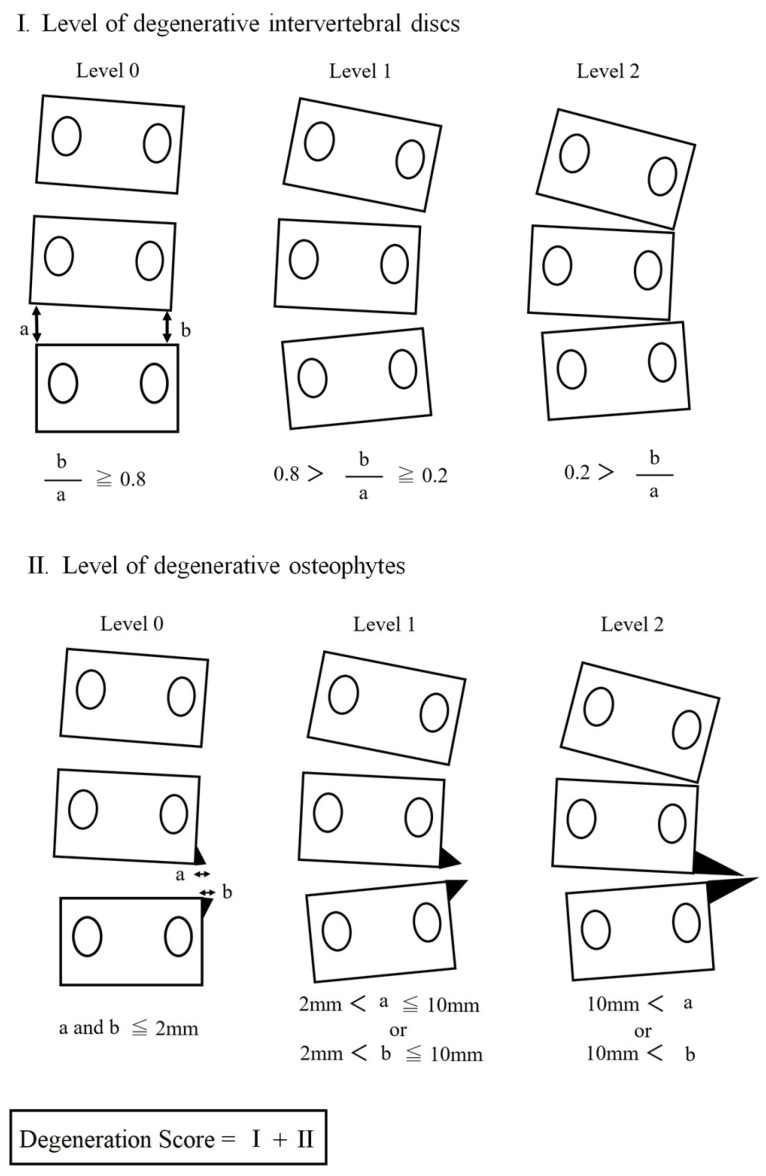
Assessment of spinal degeneration.

**Figure 2 healthcare-11-02357-f002:**
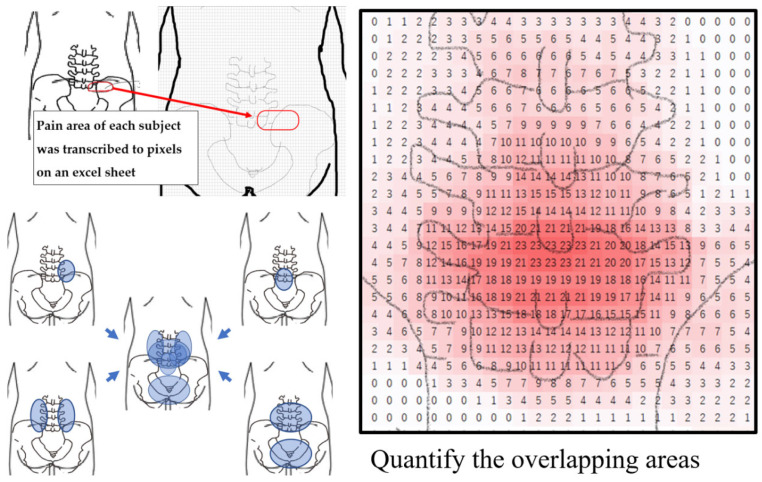
Assessment of pain area and creation of a pain heat map.

**Figure 3 healthcare-11-02357-f003:**
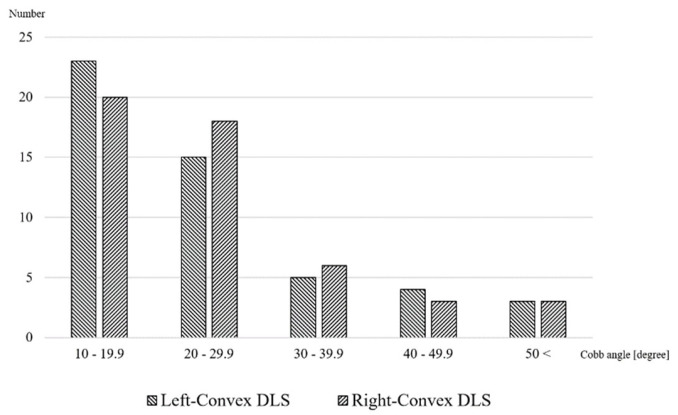
Number of participants according to Cobb angle.

**Figure 4 healthcare-11-02357-f004:**
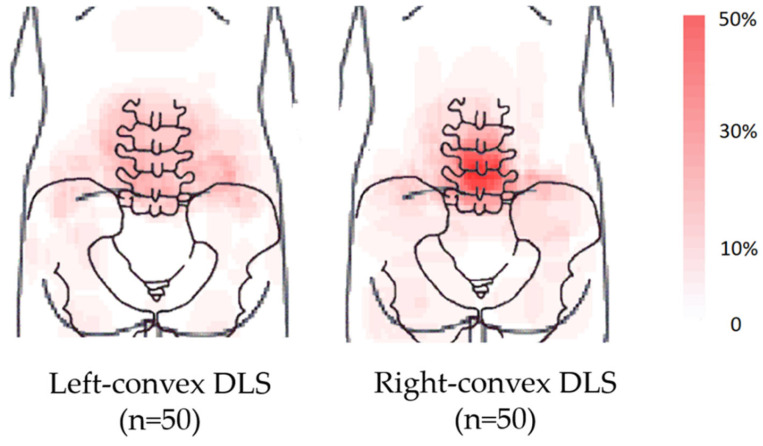
Pain heat maps according to the coronal deformity.

**Table 1 healthcare-11-02357-t001:** Comparison of variables between patients with left-DLS and right-DLS.

Variables	Lt-DLS	Rt-DLS	*p* Value
Age (yrs)	76.7 (10.3)	75.4 (6.9)	0.48
Female (%)	80.0	66.0	0.11
Pain intensity	5 [4]	4 [5]	0.90
Pain area (AU)	95.3 [132.3]	147.1 [279.3]	0.23

Each variable is compared across the three groups using χ^2^ test for categorical variables, or using the Mann–Whitney U test. Values: mean (standard deviation) or median [interquartile  range]. Abbreviations: AU, arbitrary unit; DLS, degenerative lumbar scoliosis.

**Table 2 healthcare-11-02357-t002:** Radiographic characteristics in patients with left-DLS and right-DLS.

Variables	Lt DLS (n = 50)	Rt DLS (n = 50)	*p* Value
Apical vertebrae			
L1 (n)	13	8	0.26
L2 (n)	15	20	
L3 (n)	20	22	
L4 (n)	2	0	
Degeneration score	2 [1.8]	2 [1]	0.91
Cobb angle (degree)	21.0 [12.8]	22.6 [14.9]	0.80
Major pain site			
Convex side dominant (n)	15	18	0.67
Center dominant (n)	17	18	
Concave side dominant (n)	18	14	

Each variable is compared across the three groups using χ^2^ test for categorical variables, or using the Mann–Whitney U test. Values: number (n) or median [interquartile  range].

**Table 3 healthcare-11-02357-t003:** Multivariate regression model of factors correlated with pain intensity and pain area.

Variables	Pain Intensity		Pain Area	
	β	*p*-Value	β	*p*-Value
Age	<0.01	0.92	**−0.03**	**<0.01**
Sex	0.02 (female)	0.87	−0.07 (female)	0.41
Curve side	0.01 (Lt), 0.00 (Rt) #	0.92	**−0.45 (Lt)**, 0.00 (Rt) #	**0.03**
Cobb angle	<0.01	0.36	−0.01	0.21

# Reference. Bold numbers denote statistical significance.

## Data Availability

The data used in this study are available from the corresponding author upon request.

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
