# Peer review of "Associations between Degenerative Lumbar Scoliosis Structures and Pain Distribution in Adults with Chronic Low Back Pain"

_healthcare, 2023, doi:10.3390/healthcare11162357_

Round 1
Reviewer 1 Report (New Reviewer)
The authors present an interesting study although their primary hypothesis could unfortunately not be confirmed. Nonetheless, publication of these negative data ist important and sheds further light on these chronic back pain syndromes.
Minor editing (punctuation etc.)
Author Response
Reviewer 1:The authors present an interesting study although their primary hypothesis could unfortunately not be confirmed. Nonetheless, publication of these negative data ist important and sheds further light on these chronic back pain syndromes.
Comments on the Quality of English Language, Minor editing (punctuation etc.)
Reply:
Thank yu for your comments.
We have revised the manuscript..

Reviewer 2 Report (New Reviewer)
Dear Authors,
This paper discusses interesting research area regarding the chronic low back pain in older adults. The study focuses, whether degenerative lumbar scoliosis and deformities in frontal (coronal) plane associate with the pain. The paper is well structured and well written, but some recommendations could improve its quality. I hope my comments and minor revisions would be helpful for improvement of the manuscript, before the manuscript could be considered for publication. Please find below my comments for the paper.
-Title:
-Instead of Relationships, it would recommend to use Associations.
-Instead of distributions, better to use distribution.
-I would suggest from the title of the paper to let readers know, about the age of adults is the study. So would recommend instead of just adults to use “older adults” or “elderly”.
-Keywords: add elderly (or older adults) –the same, how it will be in the title.
-In Materials and Methods, in Participants:
There should be mentioned about the age of participants, so it would be clear, that the study was focused on elderly/older adults.
Author Response
Reviewer 2:This paper discusses interesting research area regarding the chronic low back pain in older adults. The study focuses, whether degenerative lumbar scoliosis and deformities in frontal (coronal) plane associate with the pain. The paper is well structured and well written, but some recommendations could improve its quality. I hope my comments and minor revisions would be helpful for improvement of the manuscript, before the manuscript could be considered for publication. Please find below my comments for the paper.
-Title:
-Instead of Relationships, it would recommend to use Associations.
-Instead of distributions, better to use distribution.
-I would suggest from the title of the paper to let readers know, about the age of adults is the study. So would recommend instead of just adults to use “older adults” or “elderly”.
Reply:
Thank you for your suggestions.
We have revised some words as you suggested, however, we have not changed ‘adults’, because the inclusion criteria for this study was age ≥ 40 years.
-Keywords: add elderly (or older adults) –the same, how it will be in the title.
Reply:
Thank you for your suggestions, as you indicated, 90% (90/100) of the study subjects were older than 65 years old, but the remaining 10 do not meet the definition of elderly, so we do not agree with this proposal.
-In Materials and Methods, in Participants:
There should be mentioned about the age of participants, so it would be clear, that the study was focused on elderly/older adults.
Reply:
Thank you for your suggestions, as you indicated, 90% (90/100) of the study subjects were older than 65 years old, but the remaining 10 do not meet the definition of elderly, so we do not agree with this proposal.

Reviewer 3 Report (New Reviewer)
The article entitled “Relationships Between Degenerative Lumbar Scoliosis Structures and Pain Distributions in Adults with Chronic Low Back Pain” aims to investigate the location and distribution of pain in adults with chronic low back pain (LBP) with or without degenerative lumbar scoliosis (DLS) according to coronal deformities. The authors intend to relate the distribution of low back pain in subjects with degenerative lumbar scoliosis with the side of the curve where they report greater low back pain. The introduction is consistent with the approach of the study, although it is necessary to improve the justification and relevance of the study. The methodology is correct, although possibly the most pressing problem is the use of a method for quantifying the distribution of low back pain without proven reliability. Finally, and for a better evaluation of the paper by the Editorial Board and the external reviewers, authors are requested to send a final document without change control on subsequent occasions. However, this reviewer proposes the following suggestions for improving the methodological quality of the article:
ABSTRACT
The objective of the study described in this section (Ln 57-58) does not correspond to that described in the introduction section (Ln 239-240). At the beginning the authors talk about subjects with and without DLS, while later they talk about adults with CLBP and DLS. Please unify criteria because it is a crucial aspect of the study.
INTRODUCTION
The justification of the study should be improved. The authors hypothesize that the subjects must have pain on the side of the convexity, when they have previously suggested reasons why this pain could appear on both sides of the curve.
METHODS
Has the study protocol been previously registered in any database? If not, detail in the document.
What “withdrawal strategy” are the authors referring to? It would be advisable to clarify it in the document.
Regarding the inclusion criteria, why include all subjects with CLBP, and not include a criterion that refers to lumbar degeneration and the degree of scoliosis?
Likewise, and with a view to a better interpretation of the results, it would have been advisable to include a control group represented by subjects with non-degenerative idiopathic scoliosis (as the authors began by describing in lines 57-58).
Is the assessment of degeneration (subheading 2.6) based on any previous publication? Please provide reference.
Likewise, provide references to studies that have previously used this method of creating and quantifying pixels from the body chart filled out by the subject according to their pain. If not, provide data on the reliability of the method.
RESULTS
In tables 1 and 2, three groups are written when there should be two. Detail in the legend the meaning of Lt-DLS and Rt-DLS.
The authors found a relationship between pain severity and right-sided scoliosis. Could it be related to the fact that a greater proportion of the subjects had their dominant side being the right? Was this data verified?
Author Response
Reviewer 3:The article entitled “Relationships Between Degenerative Lumbar Scoliosis Structures and Pain Distributions in Adults with Chronic Low Back Pain” aims to investigate the location and distribution of pain in adults with chronic low back pain (LBP) with or without degenerative lumbar scoliosis (DLS) according to coronal deformities. The authors intend to relate the distribution of low back pain in subjects with degenerative lumbar scoliosis with the side of the curve where they report greater low back pain. The introduction is consistent with the approach of the study, although it is necessary to improve the justification and relevance of the study. The methodology is correct, although possibly the most pressing problem is the use of a method for quantifying the distribution of low back pain without proven reliability. Finally, and for a better evaluation of the paper by the Editorial Board and the external reviewers, authors are requested to send a final document without change control on subsequent occasions. However, this reviewer proposes the following suggestions for improving the methodological quality of the article:
ABSTRACT
The objective of the study described in this section (Ln 57-58) does not correspond to that described in the introduction section (Ln 239-240). At the beginning the authors talk about subjects with and without DLS, while later they talk about adults with CLBP and DLS. Please unify criteria because it is a crucial aspect of the study.
Reply:
Thank you for pointing out our mistake. As you pointed out, there is a discrepancy between the abstract and the introduction in terms of content. In the previous paper, we included CLBP patients without scoliosis as controls, and it was pointed out that this was not appropriate for the purpose of this paper, so we omitted the controls in this one.
We have corrected the part you pointed out.
INTRODUCTION
The justification of the study should be improved. The authors hypothesize that the subjects must have pain on the side of the convexity, when they have previously suggested reasons why this pain could appear on both sides of the curve.
Reply:
The third paragraph has been revised to logically explain the justification for the purpose of this study.
METHODS
Has the study protocol been previously registered in any database? If not, detail in the document.
Reply:
Since the protocol for this study is new to us, we describe it in detail in the Methods section.
What “withdrawal strategy” are the authors referring to? It would be advisable to clarify it in the document.
Reply:
The term was changed to "opt-out strategy" to clarify the term and the relevant citation was added.
Regarding the inclusion criteria, why include all subjects with CLBP, and not include a criterion that refers to lumbar degeneration and the degree of scoliosis?
Reply:
Thank you very much for your kind remarks. We have followed the reviewer's instructions and made the correction.
Likewise, and with a view to a better interpretation of the results, it would have been advisable to include a control group represented by subjects with non-degenerative idiopathic scoliosis (as the authors began by describing in lines 57-58).
Reply:
As mentioned above, in the previous paper, we included CLBP patients without scoliosis as controls. We have corrected the part you pointed out.
Is the assessment of degeneration (subheading 2.6) based on any previous publication? Please provide reference.
Reply:
In this study, we utilized a new assessment for lumbar degeneration in order to determine whether a participant's lumbar scoliosis can be regarded as DLS, and to identify differences in degeneration severity between right-convex and left-convex DLS. This assessment method has not been reported previously. However, we believe that this method is valid for the assessment of degeneration severity in the lumbar spine.
Likewise, provide references to studies that have previously used this method of creating and quantifying pixels from the body chart filled out by the subject according to their pain. If not, provide data on the reliability of the method.
Reply:
As with the above, this method was conceived by us and has not been reported previously.
The researcher transcribed each individual's pain site from an A4 template sheet to an Excel sheet. In doing so, the pain site was reproduced as much as possible. For reliability evaluation, two researchers checked the position of the transcription.
RESULTS
In tables 1 and 2, three groups are written when there should be two. Detail in the legend the meaning of Lt-DLS and Rt-DLS.
Reply:
Thank you for pointing out our mistake. We have corrected the part you pointed out.
The authors found a relationship between pain severity and right-sided scoliosis. Could it be related to the fact that a greater proportion of the subjects had their dominant side being the right? Was this data verified?
Reply:
The reviewer points out that "the authors found a relationship between the degree of pain and right scoliosis." We believe there is a misunderstanding in this point. Are you talking about area?
As shown in Table 1, there is no difference in pain intensity between the right and left convex groups. Also, it is true that the table3 shows a # for the right convex group, but this is the criterion, and the partial regression coefficient shows how much the left convex group changes relative to this criterion.
Regarding the latter point:
As shown in Table 2, those who complain of right-sided predominant pain are left convex concave pain (18) + left convex convex pain (18), 36 in total. Those who complained of left-sided predominant pain were: left convex convex side pain (15) + left convex concave side pain (14), 29 participants in total. As noted, a higher percentage of participants tended to report pain on the right side.

Round 2
Reviewer 3 Report (New Reviewer)
The authors have done a proper job of reviewing and improving the article.
This manuscript is a resubmission of an earlier submission. The following is a list of the peer review reports and author responses from that submission.
Round 1
Reviewer 1 Report
This study aimed to investigate the location and distribution of pain in
adults with chronic low back pain with or without degenerative lumbar scoliosis (DLS) according to coronal deformities. These are my comments and concerns:
Introduction:
Introduction is nicely written with the clear aim of the study.
Methods:
Were your patients similar in other characteristics - for example, work conditions, BMI, sports and activity level, duration of their symptoms, existence of other spinal pathology? This could have influence results.
Please, include information for the software used (Image-J®). Did you check the normality of the data and how?
Results:
Baseline demographic characteristics you gathered are quite limited (regarding that LBP is multidimensional phenomenon and various factors can have effect on LBP). Several demographic factors you did not collect could present serious bias for the sample of this size.
Discussion:
As you mentioned in limitation paragraph - other reasons could have contributed to the results. I believe your sample size is too small for the conclusion you made without taking into account other factors.
The scientific contribution of this study is limited.
Reviewer 2 Report
The article being formatted already does not lend itself to readability.
Abstract: Participants are divided into 3 groups then later it seems like comparisons are only made between 2 groups. This is inconsistent.
There is not enough background given to support the research question.
Some terms (de novo DLS [line 34], destructive facet joints [48]) should be updated to be readily understandable.
Line 51-56: How are you defining pain location and distribution for the purposes of this study? How did previous studies define these same ideas? Give more detail regarding other studies that observed hip and knee pain to give more context to the current study.
Line 59: change from “and data were obtained from medical records” to “retrospective review of medical records.”
Line 73: Earlier, data is said to have come from medical records review, and that is not what is described here.
Line 89: not appropriate to restate hypotheses at this point. This should come at the end of the introduction.
Line 98: with a picture of only the low back it is impossible for participants to mark every pain area on the body.
Where are definitions for right, central, and left sides of the body?
With the low back being notorious for lack of specificity of sensation, a self-report measure with pictures seems unreliable at best.
Line 112: Look into more recent literature.
Line 116: “intervention and non-intervention” terms are not consistent with what has been described up to this point in the manuscript.
Statistical methods do not align well with questions under consideration.
Line 131: describe significance in terms of alpha rather than p.
Line 140: which test was used for which comparison? Why were so many comparisons run? What sort of multiple comparison adjustments were made? Needs more information.
Lines 140-144 are repetitive with what has previously been stated.
Line 161: Where did the Spearman come from?
Table 3: I suspect that with an appropriate multiple comparison adjustment (e.g. Bonferroni) the curve side and pain area would not be significant either.
Line 171: replace “trend” with “difference.”
Line 190: remove “and not specific.”
Paragraph starting at line 191 is more of an introductory paragraph.
Line 202: “controlling confounders,” what exactly did you control and how? Are you referencing inclusion/exclusion criteria?
Sentence beginning on line 203 is not supported by the data presented.
All limitations should be listed together in the limitations section/paragraph.
The majority of literature used is far too old to support the research and the ideas presented here. It needs to be updated to show current need in this area.